# Lung and Heart Biology of the Dp16 Mouse Model of down Syndrome: Implications for Studying Cardiopulmonary Disease

**DOI:** 10.3390/genes14091819

**Published:** 2023-09-19

**Authors:** Kelley L. Colvin, Kathleen Nguyen, Katie L. Boncella, Desiree M. Goodman, Robert J. Elliott, Julie W. Harral, Jill Bilodeaux, Bradford J. Smith, Michael E. Yeager

**Affiliations:** 1Linda Crnic Institute for Down Syndrome, University of Colorado, Aurora, CO 80045, USAdesiree.goodman@cunaschutz.edu (D.M.G.); 2Department of Bioengineering, University of Colorado Denver, Anschutz Medical Campus, Aurora, CO 80045, USA; knguyen@meddux.com (K.N.); katie.boncella@cuanschutz.edu (K.L.B.); robert.elliott@cuanschutz.edu (R.J.E.); jill.bilodeaux@cuanschutz.edu (J.B.); bradford.smith@cuanschutz.edu (B.J.S.); 3Department of Medicine, University of Colorado, Aurora, CO 80045, USA; julie.harral@cuanschutz.edu; 4Section of Pediatric Pulmonary and Sleep Medicine, Department of Pediatrics, School of Medicine, University of Colorado, Aurora, CO 80045, USA

**Keywords:** respiratory tract infection, Down syndrome, mouse model, interferons, bronchus-associated lymphoid tissue

## Abstract

(1) Background: We sought to investigate the baseline lung and heart biology of the Dp16 mouse model of Down syndrome (DS) as a prelude to the investigation of recurrent respiratory tract infection. (2) Methods: In controls vs. Dp16 mice, we compared peripheral blood cell and plasma analytes. We examined baseline gene expression in lungs and hearts for key parameters related to susceptibility of lung infection. We investigated lung and heart protein expression and performed lung morphometry. Finally, and for the first time each in a model of DS, we performed pulmonary function testing and a hemodynamic assessment of cardiac function. (3) Results: Dp16 mice circulate unique blood plasma cytokines and chemokines. Dp16 mouse lungs over-express the mRNA of triplicated genes, but not necessarily corresponding proteins. We found a sex-specific decrease in the protein expression of interferon α receptors, yet an increased signal transducer and activator of transcription (STAT)-3 and phospho-STAT3. Platelet-activating factor receptor protein was not elevated in Dp16 mice. The lungs of Dp16 mice showed increased stiffness and mean linear intercept and contained bronchus-associated lymphoid tissue. The heart ventricles of Dp16 mice displayed hypotonicity. Finally, Dp16 mice required more ketamine to achieve an anesthetized state. (4) Conclusions: The Dp16 mouse model of DS displays key aspects of lung heart biology akin to people with DS. As such, it has the potential to be an extremely valuable model of recurrent severe respiratory tract infection in DS.

## 1. Introduction

With an incidence of one in 700–1000 live births worldwide, Down syndrome (DS), or trisomy of human chromosome 21 (Hsa21), is the most common chromosomal abnormality. The greatest cause of morbidity and mortality worldwide in people with DS is respiratory tract infection (RTI) [1]. As is the case in the typical population, most influenza-related deaths of individuals with DS are due to secondary bacterial infection with *S. pneumoniae* and other bacteria [2]. These so-called superinfections are clinically significant for persons with DS, as evidenced by higher rates of morbidity and mortality during the recent pandemics in 2009 of influenza (H1N1) and in 2020 of SARS-CoV-2 [3].

RTI is the most serious life-threatening health problem for persons with DS, regardless of age [1,4,5,6]. Unfortunately, the mechanisms responsible for increased lung infection mortality in DS are as yet poorly defined. Preclinical models of DS have been instrumental in furthering our understanding of the unique biology of persons with DS [7,8]. In the last several years, newer mouse models of DS have been constructed that duplicate some, but not all, of the mouse genes that are syntenic for human chromosome 21. Our institute recently found that trisomy 21 is associated with a variable over-expression of several members of the interferon signaling family [9]. Interestingly, the Dp(16Lipi-Zfp295)1Yey mouse model of DS (hereafter referred to as Dp16), which has a segmental duplication of genes on mouse chromosome 16 that are syntenic with human chromosome 21 [10], also displays aspects of deranged interferon biology [11].

Given the importance of interferon signaling in both the susceptibility and response to lung infection, we examined the basic cardiopulmonary biology and interferon gene expression of Dp16 mice as a prelude to future investigations involving viral and bacterial pathogens. We report here that the lungs of the Dp16 mouse model of DS display some elements of activated interferon expression and signaling, have increased elastance and enlarged airspaces, and contain follicularized bronchus-associated lymphoid tissue. Collectively, these features may predispose Dp16 mice to severe lung infection. Additionally, Dp16 mice circulate a unique blood plasma cytokine profile and show decreased cardiac contractility. Thus, the Dp16 mouse recapitulates many salient features of the complex lung-heart biology of DS.

## 2. Materials and Methods

### 2.1. Mice

Dp(16Lipi-Zfp295)1Yey (abbreviated Dp16) mice (Strain # 013530, The Jackson Laboratory) were housed in pathogen-free rooms under a 12 h light/dark cycle photo period. The mice were provided with Teklad Global Soy Protein-Free Extruded Rodent Diet Irradiated (2920X) and sterilized water ad libitum. All mice were treated according to the Guide for the Care and Use of Laboratory Animals, Animal Welfare Act, and PHS Policy. Littermates (C57Bl/6J strain background) lacking segmental duplication of chromosome 16 were used as controls. For euthanizing mice, euthanasia solution (390 mg pentobarbital sodium, 50 mg phenytoin sodium, per mL, VetOne, Inc., Boise, ID, USA) was injected (intraperitoneal) at 270 mg/kg. Mice were placed in a weigh boat for mass measurement. After ataxia passed, the mice were then placed in a prone position with their limbs maintained in a natural posture. The mice were measured from tip of the nose to the posterior section where the tail begins in centimeters using a standard ruler. Time until non-responsiveness was measured by starting a timer at injection of ketamine/xylazine (see Flexivent methods) and concluding when the mice did not respond to any stimulus such as ocular touch or a toe pinch.

### 2.2. RNA Sequencing

Immediately following sacrifice, 40 mg samples of homogenized lung parenchyma were randomly selected for RNA extraction and placed on ice. Total RNA was extracted from the inferior lobe of the right lung in mice (males: 3Dp16, 3 controls, females: 3Dp16, 3 controls) with a mean age of 11.5 weeks (range: 10–15 w) using Qiagen RNeasy Mini Kit (Cat. No. 74106). This study size was selected as a balance between reasonable power (β = 0.35, *p* < 0.05) and cost and practicality. Furthermore, this design followed published recommendations for RNA sequencing experiments to include at least six replicates per group (6 controls, 6 Dp16) [12]. RNA quality was assessed using Agilent Tape Station 4200. The libraries were constructed using Tecan Universal Plus mRNA-SEQ with NuQuant (Cat No. 0520). Sequencing was performed using Illumina NovaSEQ6000 Instrument 2 × 150 Paired End sequencing. We used the package DESeq2 to perform analyses of differentially expressed genes (DEG) using default parameters [13]. DEG were scored as a fold change > 1.5 vs. controls and a relaxed threshold of *p* < 0.05.

### 2.3. Reverse-Transcriptase Semi-Quantitative Polymerase Chain Reaction (RTqPCR)

Expression of transcripts for interferon signaling genes was measured by quantitative reverse-transcriptase polymerase chain reaction (qPCR). Pre-validated Taqman Assays (ThermoFisher) were used to assess transcripts: IFNAR1 (Mm00439544_m1), IFNAR2 (Mm00494916_m1), IFNGR2 (Mm00492626_m1), IL10RB (Mm00434157_m1), IRF7 (Mm00516793_g1), USP18 (Mm01188805_m1), and MX1 (Mm00487796_m1). Conditions used to perform complementary DNA transcription, polymerase chain reaction cycling, and normalization were performed as we previously described [14].

### 2.4. Hemodynamics: Pressure-Volume Measurements

Animals were anesthetized with 4–5% isoflurane in room air at 1 lpm for 2 min, a tracheal cannula was placed via cut-down and connected to a Hallowell EMC Microvent I. Anesthesia was maintained at 1.5–2.5% isoflurane in 21% oxygen/ balance nitrogen. Peak airway pressure was maintained between 10 and 12 cmH_2_O, breaths per minute at 50–75, and the oxygen flow at 0.5 lpm. The RV, PA, and LV pressures were directly measured with a 1.2-Fr Pressure-Volume Catheter (Transonic/Scisense, FTH-1212B-3518) inserted into the heart directly through the chest wall via a diaphragmatic approach. The pericardium was resected and a small hole was made at the apex of the RV with a 30-gauge needle. The pressure-volume catheter was inserted in the puncture and advanced along the length of the RV. Steady-state hemodynamics were collected with short pauses in ventilation (up to 10 s) to eliminate ventilator artifact from the pressure-volume recordings. Occlusions of the inferior vena cava were also performed to decrease preload and obtain an accurate end-systolic pressure-volume relationship. The catheter was then inserted into the PA and pressure measurements recorded. One final puncture was made in the apex of the LV and the catheter placed along the length of the LV for measurement of steady-state data in the LV. Data were recorded continuously with LabScribe 4 (iWorx, Dover, NH, USA) and analyzed offline. At the conclusion of hemodynamic measurements, mice were euthanized. Tissues from mice used for hemodynamics were not processed for other assays.

### 2.5. Fulton Score and Gravimetric Analyses

Following euthanasia, hearts from Dp16 and control mice were excised, and the right ventricle was removed from the interventricular septum and left ventricle. Fulton scores were calculated as the right ventricle mass/(left ventricle plus interventricular septum masses). In separate experiments, following euthanasia, hearts from Dp16 and control mice were excised, and the right ventricle, the interventricular septum, and the left ventricle removed. Each section was weighed and then dried at 60 degrees Celsius for three days and reweighed. Water mass was calculated as (wet weight minus dry weight)/dry weight. Tissues from mice used for Fulton score and gravimetric analyses were not processed for other assays.

### 2.6. Flexivent

Mice were anesthetized with an intermuscular injection of 100 mg/kg ketamine and 8 mg/kg xylazine, tracheostomized with an 18 ga thin-walled cannula, and connected to a flexiVent rodent ventilator. The baseline ventilation was initiated with a tidal volume (Vt) = 10 mL/kg, respiratory rate (RR) = 150 breaths/min, and positive end expiratory pressure (PEEP) = 3 cmH_2_O. An intraperitoneal injection of 0.8 mg/kg was administered to suppress respiratory drive and a 6 min stabilization period was provided with recruitment maneuvers (RM) at 2 min intervals. Each RM consisted of a 3 s ramp from PEEP to 30 cmH_2_O and a 3 s breath hold. The measurement sequence began with an RM followed by a stepwise quasi-static pressure-volume measurement (PV loop) with 7 incremental pressure steps from 0 to 30 cmH_2_O and 7 pressure decrements from 30 to 0 cmH_2_O over a 16 s total duration. The PEEP was then set to 9 cmH_2_O, an RM was performed, and pulmonary system impedance was measured four times over a 20 s period. Each impedance (FOT) measurement consisted of 3 s volume oscillation with 13 mutually prime frequencies from 1 to 20.5 Hz. The constant phase model was fit to the impedance to determine respiratory system elastance (H), tissue damping (G), and Newtonian airway resistance (Rn) [15]. The sequence of an RM followed by impedance measurements was repeated at PEEP = 6, 3, and 0 cmH_2_O. At PEEP = 3 and 0 cmH_2_O, a single-frequency oscillation (Vt = 10 mL/kg, RR = 60 breaths/min) was also recorded, and the single compartment model was used to estimate respiratory system resistance (Rrs) and elastance (Ers). To test the lung stability, 8 additional minutes of baseline ventilation at PEEP = 0 cmH_2_O were applied with H/G/Rn and Ers/Rrs measured at one minute intervals. Finally, a 2nd quasi-static PV loop was recorded.

### 2.7. Mesoscale Discovery ELISA

Mouse plasmas were collected following centrifugation (1200× *g*, 10 min at 4 degrees Celsius) of 500 microliters of peripheral blood, snap frozen, and stored at −80 °C. Plasma dilutions were determinant upon each analyte in the MSD MULTI-SPOT Assay System. Mouse left lungs were flushed with sterile 1× PBS, homogenized in Complete Lysis Buffer (MSD MULTI-SPOT Assay System, Gaithersburg, MD, USA), spun at 10,000× *g* for 20 min at 4 degrees Celsius. Supernatants were collected, snap frozen, and stored at −80 °C. To obtain cell cultures from whole lungs, mouse left lungs were flushed with sterile 1× PBS, put through lung tissue digestion procedure, spun down at 300× *g* for 10 min at RT, passed through 100 μm and 70 μm nylon screens, and then split into plates. Multiple cell types grew until 80–90% confluency, never passed, at which time the cells were lysed with Complete Lysis Buffer (MSD MULTI-SPOT Assay System) and the adherent cell protocol per the whole cell lysate preparation instructions. To collect whole lung cell culture supernatants, 0.1% serum media was placed on mouse whole lung adherent cells for 18 h before supernatants were collected and spun at 400× *g* to remove any debris. Supernatants were used neat in the MSD MULTI-SPOT Assay System. Cytokines were assayed using the V-Plex Proinflammatory Panel 1 Kit according to manufacturer’s protocol (Mesoscale Discovery) using plasma samples diluted 50:50 into diluent. All standards and plasma samples were run in triplicate. In cases where an analyte was above or below the range of the kit standards, plasmas were run again at dilutions suitable for the dynamic range of the kit for that analyte.

### 2.8. Immunoblot

Mice lung protein lysates were prepared by addition of RIPA buffer supplemented with protease, kinase, and phosphatase inhibitors (HALT, Pierce-Thermo Scientific, Waltham, MA, USA). Twenty-five μg of sample protein per lane was electrophoresed on 4–12% gradient Bio-Tris gels (NOVEX, Invitrogen, Waltham, MA, USA) followed by transfer to PVDF membrane (NEN Life Science Products, Wolnzach, Germany) in buffer (Tris-glycine, NOVEX, Invitrogen) with 10% methanol. Pre-stained molecular mass marker proteins (Bio-Rad. Hercules, CA, USA) were used to estimate band sizes. Luciferase chemoluminescence utilized Western Blot Chemiluminescence Reagent (GE Amersham, Buckinghamshire, UK). Antibodies against glyceraldehyde-3-phosphate dehydrogenase (GAPDH) were used to probe blots and estimate loading equivalencies. Scanned images of gels were acquired (iBright CL1000, Thermo) from each blot of three independent experiments. Densitometry was performed according to manufacturer’s protocol (Thermo) using normalization by β actin protein expression (all antibodies used in the study are listed in Appendix A).

### 2.9. Immunofluorescence

Immunofluorescent staining was performed using frozen sections cut at 5 μm thickness. Sections were fixed in ice cold 1:1 acetone/methanol for 10 min, air dried, and rehydrated in PBS. Sections were blocked for 30 min at room temperature in 50% fetal bovine serum/phosphate-buffered saline (FBS/PBS). Primary and fluor-conjugated secondary antibodies were sequentially applied in 5% FBS/PBS for 1 h each. The antibodies used are listed in Appendix A. Sections were mounted in VectaShield with 4′,6-diamidino-2-phenylindole (DAPI) (Vector Laboratories, Burlingame, CA, USA), and images were acquired at room temperature using a Zeiss Axiovert S100 (Zeiss, Jena, Germany) fitted with Zeiss 20 × 0.4 numerical aperture and 10 × 0.3 numerical aperture objectives and an Axiocam camera (Zeiss). Antibody-reactive (positive) cells were counted and divided by the total number of cells (i.e., DAPI^+^) in three randomly chosen fields (×200 total magnification) for each experiment and the mean percentage of positive cells was obtained.

### 2.10. Histomorphometry

#### 2.10.1. Mean Linear Intercept

Lungs were harvested from seven Dp16 mice (three males/four females) and seven control mice (three males and four females) aged 9–13 weeks old and cut at the pulmonary artery on ice and snap frozen in OCT. Excised lungs were sectioned with a cryostat and stained with hematoxylin and eosin. The section inferior of the pulmonary artery was imaged at 20× magnification in six regions. Images were taken at random areas that lacked a large airway. The observer was blinded to the group identity of the sample. The MLI was calculated for each of these images from an ImageJ plugin [16] using 5 pixel spacing and vertical chords. The mean of these six images was calculated and a *t*-test was performed (Prism 9).

#### 2.10.2. Bronchus-Associated Lymphoid Tissue (BALT)

Lungs were harvested from seven Dp16 and seven controls (three males and four females each group) that were 9–13 weeks of age and frozen in OCT. Excised lungs were sectioned at ten microns with a cryostat and stained with hematoxylin and eosin. The section inferior to the pulmonary artery was examined at 10× magnification for BALT. BALT was measured semi-automatically with a custom MATLAB program that performed intensity-based thresholding in HSV space to select regions stained by hematoxylin, filtered out objects in the resulting mask smaller than lymphocyte nuclei, morphologically closed the mask to combine close areas, filled holes in the mask, filtered out objects too small to be BALT, and finally allowed for user correction of the mask before area and perimeter was reported.

### 2.11. Statistics

For comparison between two groups, student’s T test was used, with significance selected as *p* < 0.05 (Prism 9). For multiple comparisons, an ordinary one-way ANOVA was used, a Gaussian distribution of residuals was assumed, equal standard deviations were assumed, and significance was selected as *p* < 0.05.

## 3. Results

The numbers of Dp16 mice and littermate controls used in this study for each experimental endpoint are presented in Table 1. Not all mice are represented in each category since some endpoints are terminal procedures. Notably, only Dp16 mice displayed hydrocephaly, which was significantly more pronounced in females (12/56, 21.43%) compared to males (5/51, 9.80%).

There were no differences observed in the numbers of male and female offspring from littermate controls nor in the numbers of male and female offspring from Dp16 mice. Compared to the controls, Dp16 mice were shorter in length and displayed reduced body weight across the lifespan (Figure 1a). Interestingly, Dp16 mice required significantly more anesthesia per body weight compared to age-matched controls (Figure 1b). Finally, we noted that Dp16 mice had broader heads, shorter snouts, and a “stockier” body type, some features of which have been previously reported [17].

Lung tissue biopsy occurs infrequently in persons with DS. Consequently, we and others have sought to identify biomarkers from accessible sources (blood) that correlate with co-occurring conditions in DS and their progression, and with the potential to inform evidence-based clinical management [18]. Therefore, we measured a panel of proinflammatory cytokines and chemokines in the blood plasmas of the mice. In Dp16 mice vs. the controls, we found significantly higher levels of tumor necrosis (TNF)-α, interferon (IFN)-γ, keratinocyte chemoattractant/human growth regulated oncogene (KC-GRO, also known as chemokine C-X-C motif ligand 1, CXCL1), interleukin (IL)-2, IL-5, IL-6, IL-10, and IL-12p70 (Figure 2). No differences were found in plasma IL-1beta or IL-4. When comparing the controls vs. Dp16 mice with respect to sex, we noted significant increases in IL-1beta (males), IL-2 (males and females), IL-5 (females), IL-10 (males and females), IL-12p70 (females), IFN-γ (males), KC-GRO (males and females), and TNF-α (males and females) (Appendix A). These data suggest that by virtue of segmental duplication of chromosome 16, Dp16 mice circulate a unique blood plasma profile of immune mediators, as in human DS, and that such a profile may possibly predispose to suboptimal immune responses in a variety of settings.

We next evaluated whether there were differences between the lungs from the controls and Dp16 mice with regard to interferon gene signaling expression. Using RNA sequencing, we found that compared to control lungs, the lungs of Dp16 mice expressed higher levels of mRNA for the type I interferon A receptors 1 and 2 (IFNAR1 (see Figure 3a), IFNAR2), the type II interferon γ receptor 2 (IFNGR2), and the type III interferon receptor subtypes IL-10Ra and IL-10Rb (Table 2). In addition, Dp16 mice had a higher mRNA expression of several signal transducers downstream of interferon receptors, including interferon regulatory factor (IRF)-9 and interferon-stimulated gene (ISG)-15, as well as IRF-7, myxoma resistance protein (MX)-1, MX-2, and ubiquitin specific peptidase (USP)-18. No differences were noted in the expression of the suppressor of cytokine signaling (SOCS)-3. Interestingly, we observed DEG in the X chromosome in addition to the autosomes.

Additional data on differential gene expression regulating lung-heart vascular signaling, lymphatic vasculature, Toll-like receptor (TLR) signaling, leukocyte biology, extracellular matrix biology, lung epithelium, apoptosis, and proliferation are provided in Table 2. Using a subset of select genes, we validated the expression differences we had observed by RNAseq for IFNAR1, IFNGR2, and IL-10Rb with RTqPCR performed on aliquots of RNA isolated in parallel at the time of sacrifice (Figure 3b). Interestingly, the differences we observed in interferon gene mRNA expression between the controls and Dp16 mice were not consistently observed at the protein level and showed considerable variability (Figure 3c). We found decreased IFNAR1 protein in the lungs of male Dp16 mice compared to either the controls or to female Dp16 mice (Figure 3c). We were unable to reliably detect IFNGR2 protein in the lungs of mice despite using three commercially available antibodies. We found no difference in lung IL-10Rb protein expression in Dp16 mice compared to the controls. We found increased total signal transducer and activator of transcription (STAT)-3 in female Dp16 mice compared to the controls, and increased phosphorylated STAT-3 (pSTAT3) in male and female Dp16 mice vs. the controls (Figure 3c). STAT proteins transduce interferon receptor signals and have been previously shown to be increased in tissues from people with DS [19] and from Dp16 mice tissue [20]. We found no difference in the total STAT-1 or phosphorylated STAT-1 in the lungs of Dp16 mice versus the controls. Consistent with the elevated mRNA level of USP18 in Dp16 mice vs. the controls, the protein levels of USP18 were significantly elevated in Dp16 mice lungs (Appendix A).

Given the risk of lethal pulmonary infection in DS, we evaluated the expression of platelet activating factor receptor (PAFR) because studies have suggested that an increased expression of PAFR in mouse lung epithelium enhances susceptibility to lung infection through increased bacterial adhesion [21]. Notably, the relationship between increased lung PAFR protein expression and augmented bacterial adhesion has recently been challenged [22]. We found no differences in lung PAFR protein expression in the lungs of female Dp16 mice vs. the controls, and significantly decreased lung PAFR in male Dp16 mice vs. the controls (Figure 3d). These data suggest that Dp16 mice would be unlikely to have increased PAFR-mediated bacterial adhesion, at least at initial exposure.

To begin to gauge the putative contribution of the Dp16 lung to the unique inflammatory profile we found in the blood, we used the same multi-ELISA to measure protein expression and secretion from the whole lung, lung cell culture, and the supernatants from lung cell culture. Among those same proteins we had examined in blood plasma, we found differences in expression between primary mouse lung cell cultures from the control mice vs. Dp16 mice (Table 3). Similarly, we found differences in the amount of cytokines and chemokines secreted from the lung cell cultures. Collectively, our results suggest that the inflammatory profile in the blood plasma of Dp16 mice, and putatively from persons with DS, likely does not mirror the inflammatory state of the lung and possibly other organs as well.

Persons with DS often present with pulmonary system congenital abnormalities such as cystic change, complete tracheal rings, and tracheomalacia which may contribute to increased pulmonary morbidity and mortality [23]. Pulmonary function testing in DS is complicated due, in part, to poor compliance, especially in children [24]. We therefore performed pulmonary function testing in mice using the flexivent system as published previously [25]. At all measurement onset pressures (PEEPs), we found significantly increased elastance (stiffness) in Dp16 mice compared to the controls (Figure 4a). Compared to the controls, we found no differences in airway resistance in Dp16 mice with the exception of Dp16 males at PEEP = 9 (Figure 4b). We found no differences in inspiratory capacity in Dp16 mice vs. the controls (Figure 4c). The histomorphometric analysis of mean linear intercept (MLI) provides an indirect estimate of alveolar and alveolar duct airspace size. We found that Dp16 mice showed a significant 1.3-fold increase in MLI, suggesting enlargement of the parenchymal airspaces (Figure 4d). Qualitatively, this increase appears to be primarily in the alveolar ducts. There was no evidence of pulmonary vascular remodeling. Given their role in orchestrating immune responses locally in the lung, we measured the size and cellular makeup of bronchus-associated lymphoid tissue (BALT), as we have done previously [14], and which were quite conspicuous in Dp16 mice lung sections. We found follicularized BALT in Dp16 mice compared to the controls, which were populated by CD3^+^ T cells and CD45R^+^ B cells segmented into discrete zones (Figure 4e).

Individuals with DS have high rates of congenital heart disease (CHD) and pulmonary arterial hypertension (PAH) [26]. In addition, pulmonary infection with acute respiratory distress syndrome is associated with development of pulmonary arterial hypertension [27]. We therefore examined basic cardiac function in our mice using echocardiography and invasive hemodynamic measurement. We found that most indices of left and right ventricular function were comparable between the controls and Dp16 mice (Table 4). Dp16 mice had significant decreases in right ventricular Tau-L, mean systemic pressure, systemic pulse pressure, left ventricular stroke work, and showed an increase in preload recruitable stroke work. In aggregate, these measures point to a hypotonicity of both the left and right ventricles in Dp16 mice vs. the controls. We found no inferential evidence of congenital heart disease (for example, lack of increased systemic pressure values in the right ventricle). There were no significant differences between the groups in wet to dry weight ratio, indicating that Dp16 mice do not exhibit increased myocardial water content (Figure 5a). There were no differences in right ventricular mass relative to the left ventricle (despite one control female outlier), confirming our lung histology data that Dp16 mice do not have PAH (Figure 5b). No obvious cardiac histopathology was noted (not shown). Interestingly, we found decreased levels of IFNAR1 and IL10Rb proteins in the RV of Dp16 male mice compared to Dp16 females and the controls (Figure 5c). We found no differences in the protein expression of IFNGR2 in the RV, and no differences in IFNAR1, IFNGR2, or IL-10Rb in the LVs of controls and Dp16 mice (Appendix A).

## 4. Discussion

Respiratory tract infection is arguably the most serious health burden faced by persons with DS across their lifespan. The basis for such severe lung morbidity and mortality has not yet been aggressively investigated, and lung tissues from either biopsy or autopsy from persons with DS are not widely available. The development of new and more powerful pre-clinical models of DS offers the hope for addressing this unmet clinical need. Here, we describe for the first time the unique lung and heart biology of the Dp16 mouse model of DS, as a prelude to future investigations aimed at unraveling the unique susceptibility to lung-heart diseases faced by persons with DS [26]. In aggregate, our data indicate that there are molecular (STAT3 signaling), structural (lung MLI, bronchus-associated lymphoid tissue), and physiological differences (lung elastance, cardiac hypotonicity) in Dp16 mice that may predispose them to lung and heart disease. Given these apparent similarities with trisomy 21, our work here has important clinical implications for lung and heart disease for persons with DS.

During our studies with the Dp16 mice, we found that hydrocephaly occurred in roughly 10% of the males and 20% of the females. We currently have no explanation for the sex difference. To our knowledge, there are as yet no comprehensive MRI or other studies on ventriculomegaly in DS. However, our results comport closely with Raveau et al. who reported that one third of Dp16 mice displayed severe hydrocephalus [28]. These workers were able to pinpoint a genetic locus sufficient to trigger hydrocephalus that contains genes and gene modifiers necessary for cilia function such as PCP4 [28]. PCP4 over-expression had been previously implicated in precocious neuronal differentiation and disturbed calcium signaling in the TgPCP4 mouse model [29,30].

Our finding of highly significant differences in the blood plasma levels of cytokines and chemokines between the controls and Dp16 mice closely mirrors previous reports in humans with DS. The high variance we observed in some blood plasma proteins (ex: IL-6, IL-10, IL-12p70) also matches results from published studies in human DS and the publicly available data from the ongoing human trisome project (http://www.trisome.org/explorer, accessed on 8 August 2023). A sampling of reported results indicates significant variability among individuals with DS *within studies* [31,32], and also order of magnitude differences in reported mean values and/or and large fold change calls for blood plasma proteins in people with DS *across studies*. Regardless, a distinctive circulating panel of inflammatory mediators implies a unique “steady-state” that may help to explain why persons with DS mount differential and often suboptimal responses to pathogens and to immunization efforts [33,34]. At this time, it is unclear why certain mediators or families of mediators might be different in DS. None of the blood plasma mediators we measured are the protein products of mouse chromosome 16 (syntenic to Hsa21) genes. However, the mere presence of an extra chromosome globally perturbs gene expression [35]. Importantly, Dp16 mice carry a segmental duplication and do not therefore recapitulate biological phenomena related to the presence of additional chromosomal architectural components such as centromeres and telomeres. Nevertheless, the pattern of cytokines and chemokines we detected in the blood of Dp16 mice implies a condition akin to infection with influenza virus, in which susceptibility to secondary (or “super”) infection by other viruses and by bacterial pathogens is high [2,36]. Blood analyses will continue to be an important source of potential biomarkers in DS given the lack of lung tissue available for study.

We chose to examine the Dp16 mouse model because of the segmental duplication of several interferon signaling genes on mouse chromosome 16 that are syntenic to Hsa21. As anticipated, we consistently found increased lung mRNA expression for these genes in Dp16 mice. However, we found significant discordance between mRNA levels and protein levels for these genes in Dp16 mouse lungs. To further emphasize this point, we found increases in MX1 mRNA expression, yet MX1 protein is a non-functional mutant in Bl6 mice strains [37], such as the Dp16 model. Thus, the extrapolation that increased MX1 mRNA implies that similar increases in *functional* MX1 protein activity in Dp16 mouse tissues are not likely. Collectively, our immunoblot data in the lung point to an activation of pSTAT3 pathways occurring in the absence of any significant increases in IFN receptor protein levels. Currently, we are unable to determine what might be over-activating STAT3 signaling in the basal state in Dp16 mice. The STAT3 pathway can be activated by a large number of factors including Toll-like receptors, interleukin receptors, g-protein-coupled receptors, microRNAs, and even mitochondria [38]. We also found considerable variability in the protein expression of segmentally duplicated genes between individual Dp16 mice (e.g., Figure 3 immunoblots). Our data raise the possibility that in trisomy 21, there is considerable person-to-person variability in Hsa21-encoded and nonHs21-encoded protein expression levels. This apparent complexity of the state of altered gene expression highlights the need for organ-specific protein studies to validate the currently available mRNA datasets. Furthermore, our findings imply that therapy design for persons with DS may greatly benefit from “personalized medicine” strategies that enhance tissue specificity for individuals and that reduce off target effects.

There are several possible mechanisms for such a heterogeneous protein expression pattern. We found increased mRNA and protein expression for ubiquitin-specific peptidase 18 (USP18, Appendix A), which, along with the suppressor of cytokine signaling (SOCS)1 and SOCS3, are negative feedback regulators of types I and III interferon signaling [39,40]. We found no increases in either SOCS1 or SOCS3 mRNA in Dp16 mice lungs vs. controls (Table 2). Our data regarding increased USP18 raises the possibility that cells and tissues can directly attenuate the consequences of Hsa21 gene (over) dosage. As mentioned, such potentially compensatory effects may not be evident if analyses are restricted to the level of RNA expression. Second, cells and tissues may reposition over-expressed interferon receptors intracellularly to avoid activation. Our tissue culture data indicate that for several protein species, intracellular expression differs both from membrane expression and from secretion. This finding supports the idea that despite an over-expression of genes at the RNA level, cells are not necessarily “compelled” to functionalize them as proteins at the cell membrane or in the extracellular space. Recent studies on cells derived from persons with DS demonstrated an increased sensitivity to type I interferon stimulation [9]. Depending on cell-type specific patterns of gene expression, cells and tissues may thus have variable capacities to modulate Hsa21 gene dosage at the effector protein level. Our finding that the lung tissue protein expression levels of chromosome 16-encoded genes were not necessarily retained by primary cell cultures grown from those same lung explants is interesting. Our current explanation is that the standard culture conditions (for example: plastic culture ware and static no-flow conditions) were inadequate to faithfully maintain the biological state of the lung tissue at the time of explant. Indeed, drastically different effects in cell and tissue cultures are obtained when utilizing biologically relevant substrates and dynamic systems that more closely resemble actual biological niches of the lung [41]. Studies are underway to determine whether cells and tissues from individuals with DS are more sensitive to culture conditions.

Lung expression of PAFR was decreased in our cohort of male Dp16 mice. PAFR is encoded on chromosome 1 in humans and chromosome 4 in mice, and therefore its expression is not directly affected by trisomy 21 in people with DS or in mice with segmental duplication of chromosome 16. Since our studies here are a prelude to investigations using infectious agents, we conclude that increased bacterial adhesion mediated through PAFR is unlikely to contribute to increased incidence of lung infection in Dp16 mice. Future studies will need to explore airway epithelial PAFR expression using lung tissue obtained from persons with DS.

Our finding of increased elastance in Dp16 mice compared to the controls is intriguing. The stiffness of the lungs (elastance) is due to the elastic recoil of the lung tissue (~1/3 contribution) and to surface tension at the air-alveoli interface (~2/3 contribution) [42]. The presence of increased MLI in Dp16 lungs suggests simplification of the parenchyma. Increased MLI would be expected to reduce the stiffness of the lung due to the reduction in stress-bearing septa. At this time, we cannot explain why the Dp16 lungs are stiffer at low volumes, where the elastin fibers play a dominant role, but have similar lung volumes, which are more strongly influenced by the collagen fiber network. Furthermore, we cannot explain why an apparently simplified parenchyma (i.e., increased MLI) has an elevated stiffness. We found significant increases in lung mRNA for Fibulin 1 and Fibulin 2 (Table 2), which have been implicated in pulmonary fibrosis [43]. There are several gene families encoded on mouse chromosome 16 that have been shown to critically regulate lung vascular wall thickness and alveolarization during development. Claudins, and A Disintegrin and Metalloproteinase with Thrombospondin-like motifs (ADAMTS) 1 and ADAMTS 5 have all been implicated in the biology of lung stiffness and extracellular matrices [44,45,46]. To our knowledge, it is unknown whether any of these gene families are upregulated in the lungs of persons with DS and if they may play roles in regulating lung alveolar surface tension.

The recent COVID-19 pandemic has unfortunately reminded us that respiratory tract infection negatively affects the cardiovascular system, variably manifesting as pulmonary hypertension, a 17-fold increased risk for heart attack [47], myocarditis, thromboembolism, and stroke [48]. We found that, at baseline, Dp16 mouse hearts appear nearly normal physiologically. We found subtle changes suggesting muscular hypotonicity in both ventricles. Our study did not anatomically address congenital heart disease in adult Dp16 mice. Our anecdotal evidence of decreased litter sizes in Dp16 vs. the controls suggests that intrauterine fetal demise due to CHD is increased in Dp16 mouse embryos. Although we have no direct evidence of spontaneous closure of septal defects in Dp16 mice, we found no physiologic evidence of higher pressures in the pulmonary circuit that would be expected with patent atrial or ventricular septal defects. Previous studies have indicated that among segmental duplication models, only Dp16 mice, and not Dp10 and Dp17 mice, exhibit CHD [49]. The Ts65Dn mouse model (one of the most published models of DS) is trisomic for a region on mouse chromosome 16 that is syntenic to Hsa21 and also displays CHD. DS is the most common genetic basis for CHD, and affects ~50% of children with DS. It is unknown whether persons with DS with a history of CHD are more susceptible to severe lung infection or to complications of lung infection compared to persons with DS with no history of CHD. To our knowledge, our report here is the first to report cardiac muscle hypotonicity and lack of functionally patent CHD in a mouse model of DS in *adult* animals.

We found that Dp16 mice showed a significant 1.3-fold increase in MLI, suggesting an increased alveolar airspace size comparable to mild-moderate emphysematous change in humans [50]. It is unclear whether increased MLI is associated with increased susceptibility to severe lung infection, although the links between acute exacerbations in COPD patients, bacterial infection, and worsening disease progression are well established [51]. Compared to the controls, we found abundant, large, follicularized BALT in Dp16 mice which were populated by regionalized CD4 T cells and CD45RA B cells. Mouse models of lung infection point to multiple roles for BALT, which are more numerous and prominent post-infection [52]. The potential contribution of BALT to the orchestration of lung immune responses to infectious agents will be a prime focus of future studies.

There are important limitations to our study. First, the Dp16 mouse model of DS contains a segmental duplication of a portion of chromosome 16. Thus, the model is a partial trisomy with regard to gene content, and it lacks a freely segregating extra chromosome. In addition, the partial trisomy includes several mouse genes that are not present on Hsa21. Thus, it cannot be assumed that the phenotypic features of the Dp16 mouse we described herein will be observed in full human trisomy 21. Second, our RNA seq studies involved a relatively low number of biological replicates. This “non-omics” approach was intentional, as is evidenced by our heavier reliance on protein analyses of gene expression in the lungs and heart and on physiological endpoints. Third, our study was designed to compare control mice to Dp16 mice and was, therefore, underpowered to rigorously examine differentially expressed genes arising from sex differences. The lack of statistical power (and concomitant risk of type II errors) to examine sex differences is somewhat mitigated by (1) the inbred nature of the Dp16 mouse strain, (2) by our focus on chromosome 16 gene expression, particularly interferon signaling genes.

A significant strength of our study is its simple design, adhering to a “as simple as possible but no simpler” approach. As such, our findings should be readily reproducible using tools and reagents accessible to most labs. Our study clearly points to the potential utility of Dp16 mice to model the lung and heart biology of persons with DS.

## Figures and Tables

**Figure 1 genes-14-01819-f001:**
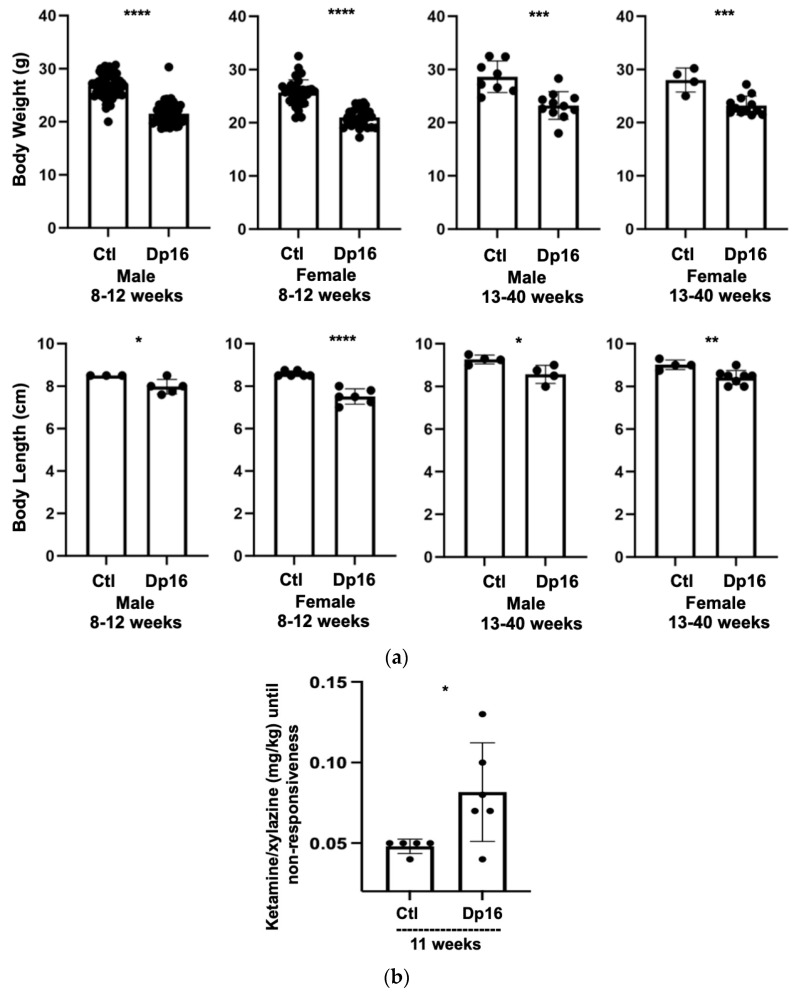
General Characteristics of control and Dp(16Lipi-Zfp295)1Yey mice. There were no differences observed in sex among offspring between littermate controls and Dp16 mice. (**a**) Compared to controls (n = 128 males, 116 females), Dp16 mice (n = 51 males, 56 females) display reduced body weight and are shorter in length (controls n = 9 males, 13 females, Dp16 n = 9 males, 14 females) across the lifespan. (**b**) In a selected cohort (n = six controls, six Dp16), Dp16 mice required more ketamine/xylazine to reach an anesthetized state. * = *p* < 0.05, ** = *p* < 0.01, *** = *p* < 0.001, **** = *p* < 0.0001.

**Figure 2 genes-14-01819-f002:**
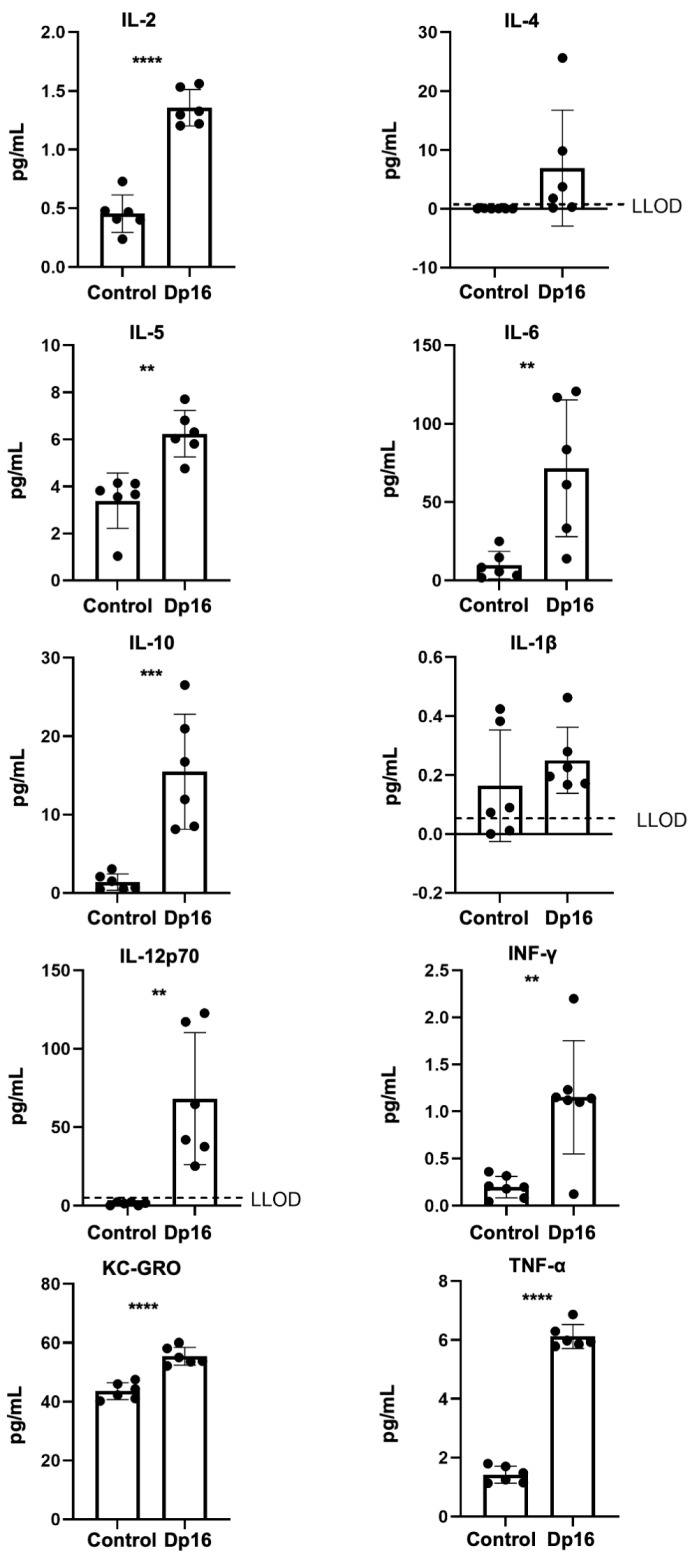
Compared to control mice (n = 6), Dp16 mice (n = 6) circulate a unique profile of blood plasma cytokines and chemokines. Higher levels were observed for tumor necrosis (TNF)-α, interferon (IFN)-γ, keratinocyte chemoattractant/human growth regulated oncogene (KC-GRO, also known as chemokine C-X-C motif ligand 1, CXCL1), interleukin (IL)-2, IL-5, IL-6, IL-10, and IL-12p70. No differences were found in plasma IL-1beta or IL-4. ** = *p* < 0.01, *** = *p* < 0.001, **** = *p* < 0.0001.

**Figure 3 genes-14-01819-f003:**
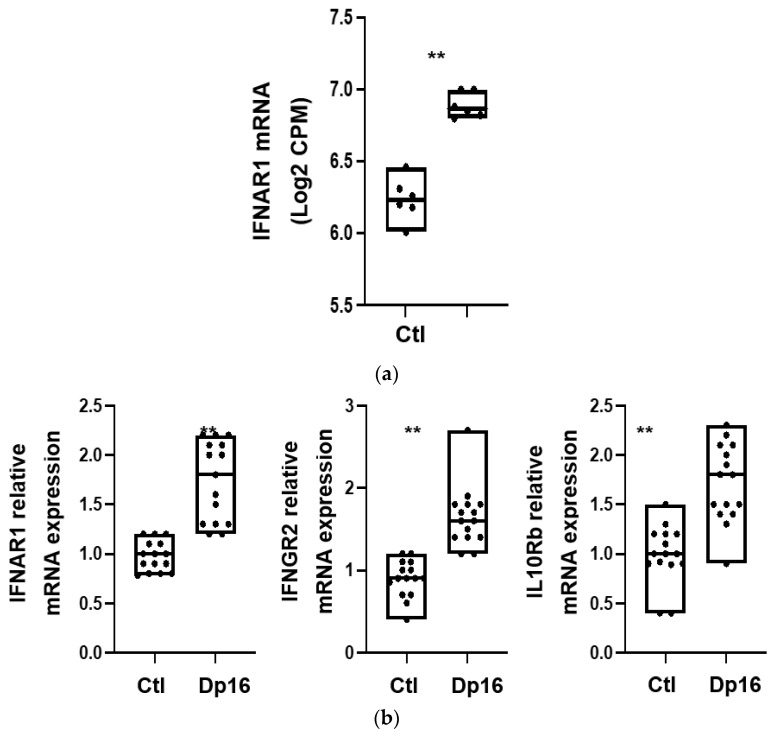
Lung gene expression differs between control mice and Dp16 mice. (**a**). Representative result from RNA seq experiment showing increased IFNAR1 mRNA in Dp16 mice lung (n = 6 controls, n = 6 Dp16). (**b**). Representative results from quantitative reverse transcriptase polymerase chain reaction showing increased mRNA levels of IFNAR1, IFNGR2, and IL-10Rb in lungs of Dp16 mice vs. controls. (n = 7 males 8 females controls, n = 7 males 8 females Dp16). (**c**). Representative IFNAR1 immunoblot, GAPDH immunoblot, and bar graphs summarizing blot densitometric data. Compared to controls, lungs from Dp16 male mice showed decreased expression of IFNAR1 protein and increased phospho-STAT3. Compared to control mice, lungs from Dp16 female mice showed increased protein expression of STAT3 and pSTAT3. IFNGR2 protein was not detectable by immunoblot in any lung samples. (n = 6 males 6 females controls, n= 6 males 6 females Dp16). (**d**). The lungs of Dp16 male mice express significantly less PAFR protein compared to controls. (n = 6 males 6 females controls, n = 6 D males 6 females p16). * = *p* < 0.05; ** = *p* < 0.01.

**Figure 4 genes-14-01819-f004:**
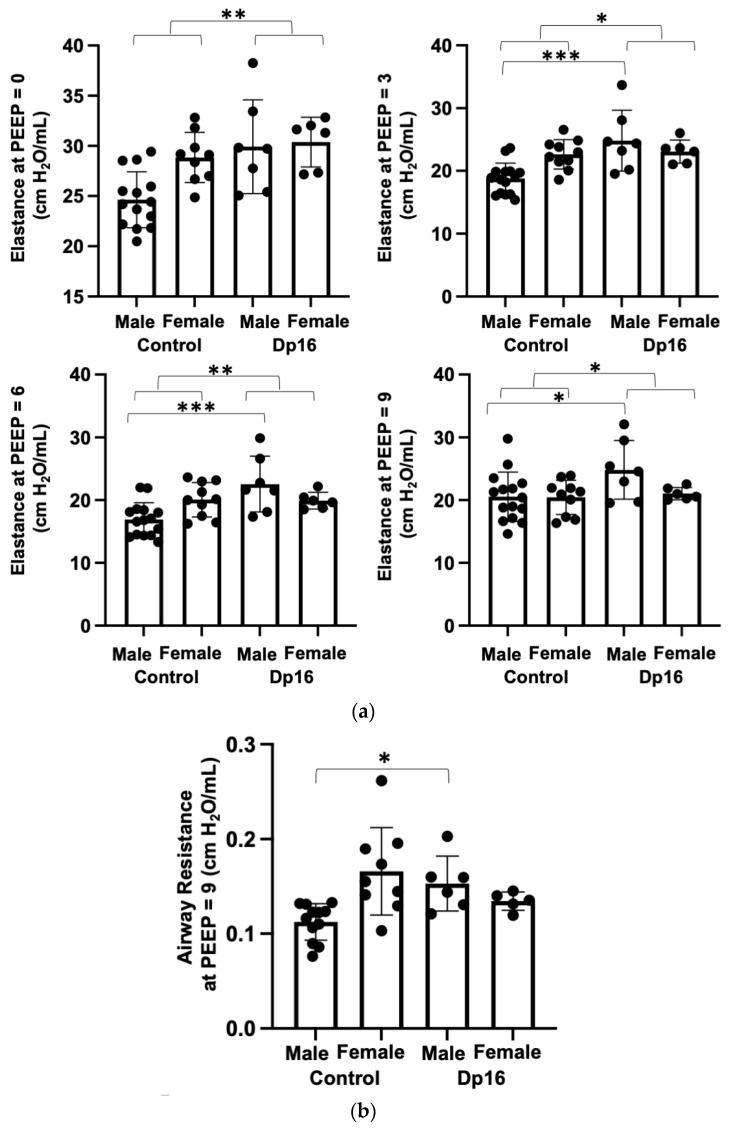
Lungs of Dp16 mice are functionally and structurally different than control mice lungs. (**a**). At all measurement onset pressures (PEEPs), elastance is significantly increased in Dp16 mice (n = 7 males 6 females) compared to controls (n = 15 males 10 females). (**b**). At (only) PEEP = 9 cm H_2_O)/mL, lung airway resistance is increased in male Dp16 mice compared to controls. (**c**). There is no significant difference in inspiratory capacity in Dp16 mice compared to controls. (**d**). MLI analysis reveals a 1.3-fold increase in mean linear intercept (MLI) of lung parenchyma in Dp16 mice compared to controls (bar graph, **left**). Representative images of hematoxylin and eosin stained sections from control (**left**) and Dp16 (**right**). Black arrows indicate regions of alveolar simplification (**e**). Compared to controls (n = 3 males 3 females), Dp16 mice (n = 3 males 3 females) display numerous follicularized bronchus-associated lymphoid tissues (left representative image of a hematoxylin and eosin stained section, black arrows) and which contain discrete zones of T cells (red) and B cells (green) (white arrows, right representative image of a immunofluoresecent stained image; blue = DAPI stained nuclei). Bar = 50 μm. * = *p* < 0.05, ** = *p* < 0.01, *** = *p* < 0.001.

**Figure 5 genes-14-01819-f005:**
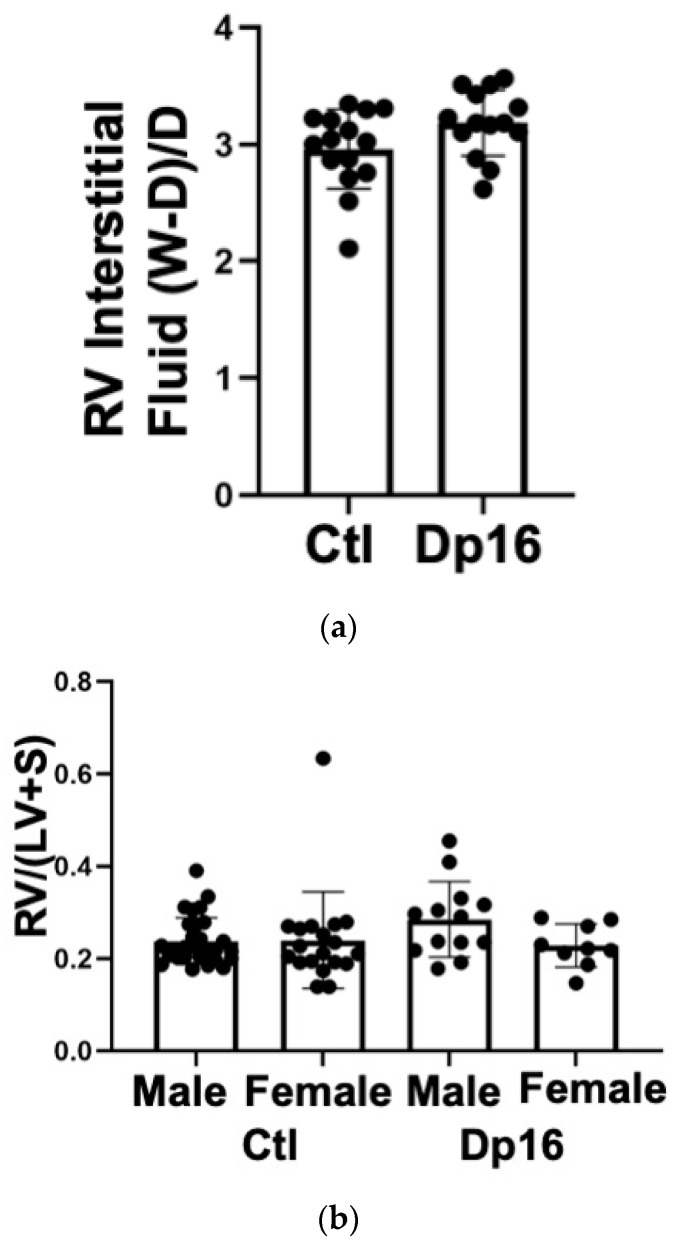
Analyses of hearts from controls and Dp16 mice. (**a**). Gravimetric data showing no differences in RV myocardial water content (wet weight minus dry weight/dry weight) in Dp16 mice (n = 7 males 7 females) vs. controls (n = 7 males 8 females). (**b**). Fulton scores (right ventricle mass/left ventricle mass plus interventricular septum mass) showing absence of right ventricular hypertrophy in Dp16 mice (n = 10 males 10 females) and controls (n = 13 males 9 females). (**c**). Bar graphs summarizing densitometric data from immunoblots of RVs from control (n = 6 males 6 females) and Dp16 (n = 6 males 6 female) mice. IFNAR1 and IL-10Rb were significantly decreased in male Dp16 mice vs. controls. * = *p* < 0.05.

**Table 1 genes-14-01819-t001:** Numbers of control and Dp(16Lipi-Zfp295)1Yey mice used in each assay in the study. There were no differences observed in the numbers of male and female offspring from littermate controls nor in male and female offspring from Dp16 mice.

Numbers of Control and Dp16 Mice Used for This Study				
Assay/Endpoint	Controls		Dp16	
	Males	Females	Males	Females
**RNA Sequencing**	3	3	3	3
**RTqPCR**	7	8	7	8
**Hemodynamics**	10	10	10	10
**Fulton Score**	10	10	13	9
**Gravimetry**	7	8	7	7
**Flexivent**	15	10	7	6
**Mesoscale ELISA-Plasma**	3	3	3	3
**Mesoscale ELISA-Cells**	3	3	3	3
**Mesoscale ELISA-Homogenates**	3	3	3	3
**Immunoblot**	6	6	6	6
**Immunofluorescence**	3	3	3	3
**Histomorphometry**	4	3	3	4
**Body weight**	128	116	51	56
**Body length**	9	13	9	14
**Anesthesia time to non-response**	3	3	3	3

**Table 2 genes-14-01819-t002:** Select lung RNA Seq data for control mice and Dp16 mice (n = 3 males and 3 females per group). The table lists names of selected genes and indicates by fold change (columns left to right), whether DEG was increased in Dp16 mice vs. controls, no difference, or decreased in Dp16 mice vs. controls. Numbers in parentheses indicate the chromosome number location of the gene. N.D. = no difference.

Gene Family	Lung mRNA Fold Change *Increased* in Dp16 vs. Controls (Chromosome Number)	Lung mRNA N.D. in Fold Change Dp16 vs. Controls (Chromosome Number)	Lung mRNA Fold Change *Decreased* in Dp16 vs. Controls (Chromosome Number)
**Type I IFN**	IFNAR1 (16) 1.6 ± 0.3 IFNAR2 (16) 1.7 ± 0.2	---	---
**Type II IFN**	IFNGR2 (16) 1.9 ± 0.1	---	---
**Type III IFN**	IL-10Ra (9) 2.4 ± 0.1 IL-10Rb (16) 1.6 ± 0.1 IRF7 (7) 2.8 ± 0.1 IRF9 (14) 1.7 ± 0.1	---	---
**IFN Activated Genes**	ISG15 (4) 2.5 ± 0.1 Mx1 (16) 1.8 ± 0.1 Mx2 (16) 1.7 ± 0.1 USP18 (6) 2.1 ± 0.1 OAS1a (5) 2.8 ± 0.1 OAS1g (5) 2.7 ± 0.1 OAS2 (5) 2.4 ± 0.1 OAS3 (5) 2.9 ± 0.1	SOCS3 (11) SOCS1 (16)	---
**Vascular Signaling**	Flt3 (5) 3.8 ± 0.1	ACVRL1 (5) Flt1 (5) PECAM (5) VECAM (5) VEGFA (5) VEGFB (5) VEGFC (5) PDGF (5) PDFGR (5) Smad1 (8) Smad 2 (18) Smad 3 (9) Smad 4 (18) Smad 6 (9) Smad 7 (18) Smad 9 (3) Tek (4) TGFB1 (7) TGFBR1 (4) TGFBRII (9) Tie1 (4)	Smad 5 (13) 2.2 ±0.1 VEGFD (X) 1.9 ± 0.1
**Lymphatic**	---	FoxC2 (8) Podoplanin (4) Prox1 (1) Sox18 (2)	---
**TLR Signaling**	IRF5 (6) 1.8 ± 0.1 TLR2 (3) 2.4 ± 0.1 TLR7 (X) 2.7 ± 0.1 TLR13 (X) 1.8 ± 0.1	---	---
**Leukocyte Biology**	Beta2M (2) 2.4 ± 0.1 CD54/ICAM (9) 2.4 ± 0.1 CD68 (11) 1.9 ± 0.1 CD86 (16) 1.7 ± 0.1 CD152/CTLA4 (1) 2.2 ± 0.1	CD28 (1) CD40 (2) CD83 (13) CD163 (6)	---
**ECM**	Fibulin 1 (15) 2.4 ± 0.1 Fibulin 2 (6) 2.1 ± 0.1	Col1a1 (11) Col1a2 (6) Col3a1 (1) Col4a1 (8) Col5a1 (2) Col18a1 (10) TCF21 (10)	---
**Lung Epithelium**	---	PAFR (4)	---
**Apoptosis**	---	Bad (19) Bax (7) Bcl2 (11)	---

**Table 3 genes-14-01819-t003:** Lung protein expression of inflammatory mediators differs between controls and Dp16 mice and differs from levels of the same inflammatory mediators in blood plasma. Table showing data from (left to right): lungs, primary cell cultures from lungs, and from supernatants from primary lung cell cultures (data in each cell represent the mean of n = 6 controls, 6 Dp16). Lung lysates showed increased expression of IL-10 in Dp16 mice vs. controls. Primary lung explant cell cultures from Dp16 mice vs. controls showed decreased expression of IL-4, IL-5, IL-6, IL-10, IL12p70, and KC-GRO, and showed increased expression of TNF-α. The supernatants of primary lung explant cell cultures from Dp16 mice vs. controls showed increased IL-10 and showed decreased IL-5, Il12p70, and IFN-γ. SD = standard deviation. bold = *p* < 0.05.

Analyte	Cell Lysate				Lung Lysate				Supernatant			
	Control		Dp16		Control		Dp16		Control		Dp16	
	Avg	SD	Avg	SD	Avg	SD	Avg	SD	Avg	SD	Avg	SD
IL-2	1.53	0.72	1.06	0.12	7.48	1.40	6.92	3.21	1.77	0.17	1.69	0.21
IL-4	**0.29**	**0.07**	**0.20**	**0.02**	1.51	0.27	2.32	1.38	0.03	0.01	0	0
IL-5	**1.05**	**0.53**	**0.31**	**0.13**	2.99	0.50	8.69	1.30	**0.28**	**0.04**	**0.08**	**0.01**
IL-6	**7180**	**1120**	**3870**	**665**	134	8.36	147	45	4200	1920	3430	1340
IL-10	**1.80**	**0.21**	**1.48**	**0.16**	**7.95**	**1.06**	**17.8**	**11.7**	**0.72**	**0.23**	**2.40**	**0.67**
IL-1β	19.1	18.4	13.1	8.1	**16.7**	**4.77**	**44.8**	**35.0**	0.07	0.01	0.06	0.02
IL-12p70	**60.7**	**15.7**	**32.7**	**2.39**	164	18.7	168	22.5	**15.0**	**2.03**	**10.8**	**3.04**
IFN-γ	0.07	0.02	0.06	0.01	1.30	1.77	1.93	3.62	**0.06**	**0.02**	**0.01**	**0.01**
KC-GRO	**70.7**	**25.2**	**7.68**	**1.48**	139	85.8	174	14.4	98.7	66.4	35.9	17.0
TNF-α	**5.27**	**0.29**	**7.45**	**1.24**	**21.1**	**4.92**	**39.7**	**15.7**	0.84	0.44	0.90	0.26

**Table 4 genes-14-01819-t004:** Hearts of Dp16 mice display hypotonicity compared to controls. Table values are means ± SD. n = 20 for control group and n = 16 for Dp16 group unless noted in superscript. HR, heart rate; mPAP, mean pulmonary arterial pressure; PaPP, pulmonary artery pulse pressure; ESP, end-systolic pressure; EDP, end-diastolic pressure; ESV, end-systolic volume; EDV, end diastolic volume; SV, stroke volume; CO, cardiac output; EF, ejection fraction; Ea, arterial elastance; ESPVR/Ees, end-systolic pressure-volume relationship/arterial elastance; Ca, pulmonary arterial compliance; PVR, pulmonary vascular resistance; MSP, mean systemic Pressure; SVR, systemic vascular resistance; SW, stroke work; PRSW, preload recruitable stroke work; Tau L, diastolic time constant Logistic decay model. Bold * = *p* < 0.05 ** = *p* < 0.01.

	Controls	Dp16	*p* Value
Weight	23.27	23.27	0.406
HR (bpm)	525 ^19^	492 ^16^	0.316
mPAP (mmHg)	16.86 ^17^	14.14 ^10^	0.138
Pa PP (mmHg)	13.45 ^17^	14.09 ^10^	0.641
RV-ESP (mmHg)	30.50	30.90 ^16^	0.789
RV-EDP (mmHg)	3.09	3.04 ^16^	0.911
RV-ESV (mL)	23.34	19.72 ^16^	0.150
RV-EDV (mL)	44.14	37.16 ^16^	0.061
RV-SV (mL)	20.80	17.64 ^16^	0.134
RV-CO (mL/min)	9.91	8.45 ^16^	0.178
RV-EF %	47.64	47.16 ^16^	0.897
RV-E_a_ (mmHg/mL)	1.68	1.92 ^16^	0.255
RV-E_a_/E_es_	1.22 ^17^	1.24 ^12^	0.973
RV-SV/ESV	1.03	0.99 ^16^	0.842
PVR (mPAP/RVCO)	1.78 ^16^	2.14 ^9^	0.167
RV- C_a_ (SV/PaPP)	1.67 ^16^	1.46 ^9^	0.621
RV-dP/dt_max_ (mmHg/sec)	2213	2476 ^16^	0.306
RV-dP/dt_min_ (mmHg/sec)	−1948	−2058 ^16^	0.577
RV-SW (mJoules)	0.08	0.07 ^16^	0.544
RV-PRSW	15.83 ^17^	12.69 ^12^	0.232
**RV-Tau L (msec)**	**19.56 ^19^**	**15.34 ^15^**	**0.015 ***
RV-ESPVR (E_es)_	2.27 ^17^	1.86 ^12^	0.448
**MSP (mmHg)**	**49.15 ^12^**	**34.21 ^14^**	**0.013 ***
**Systemic PP (mmHg)**	**36.41 ^12^**	**27.74 ^14^**	**0.007 ****
SVR (MAP/LVCO)	5.44 ^10^	4.2 ^12^	0.187
LV-ESP (mmHg)	73.7 ^19^	58.49 ^15^	0.060
LV-EDP (mmHg)	6.48 ^19^	4.51 ^15^	0.140
LV-ESV (mL)	22.33 ^18^	23.57 ^14^	0.744
LV-EDV (mL)	43.26 ^18^	41.53 ^14^	0.730
LV-SV (mL)	20.93 ^18^	17.96 ^14^	0.179
LV-CO (mL/min)	11.24 ^18^	9.23 ^14^	0.101
LV-EF %	49.91 ^18^	44.11 ^14^	0.163
LV-E_a_ (mmHg/mL)	4.19 ^18^	3.61 ^14^	0.379
LV-E_a_/E_es_	0.56 ^9^	0.67 ^3^	0.529
LV-SV/ESV	1.2^18^	0.85 ^14^	0.158
LV-dP/dt_max_ (mmHg/s)	6329 ^19^	5251 ^15^	0.327
LV-dP/dt_min_ (mmHg/s)	−5108 ^19^	−3660 ^15^	0.075
**LV-SW (mJoules)**	**0.22 ^18^**	**0.15 ^14^**	**0.023 ***
**LV-PRSW**	**19.06 ^9^**	**26.98 ^3^**	**0.048 ***
LV-Tau L (ms)	14.31 ^18^	15.14 ^14^	0.486
LV-ESPVR (E_es_)	7.79 ^9^	6.05 ^3^	0.536

## Data Availability

Data are contained within the article and Appendix A.

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
