# Peer review of "Lung and Heart Biology of the Dp16 Mouse Model of down Syndrome: Implications for Studying Cardiopulmonary Disease"

_genes, 2023, doi:10.3390/genes14091819_

Round 1

Reviewer 1 Report

Colvin et al have examined molecular and physiological features of lung and heart function in the Dp16 mouse model of Down syndrome. This is an important topic because respiratory dysfunction in Down syndrome is a serious medical problem throughout the life span and infections are a significant cause of early mortality. This work is also highly relevant because little to no attention has so far been given to analysis in model systems. Understanding the abnormalities in mouse models should lead to better understanding of the human condition and potentially better therapies.

            The importance of the work aside, there are several serious problems with the manuscript. First, at least some of the experiments suffer from flawed design: the RNA sequencing experiments used only 3 mice per genotype and sex. This is too few for reliable detection of significant differences (note that sex differences in gene expression are known which precludes pooling  male and female data). It is not clear but possibly the protein measurements also used only three mice per group; high variabiliy among individuals adds to the problem of low n. Correcting this indeed requires additional experiments. Second (and more easily rectified), there is a lack of consistent information on the number of mice used in each experiment, a frequent failure to separate data by sex, and a failure to provde quantitative results in favor of mere qualitative representations. Third, the manuscript was not carefully prepared. Indeed, the Methods and Results sections read like they were written by multiple authors, each writing their own section, but no one taking care to edit and revise to make a consistent professional presentation of details and figures/tables. Much more care and attention is needed. One or more authors need to take responsibility for consistency in the text, figures and tables, and professionalism in manuscript preparation. Finally, these data and their interpretation need to be put in the context of all availble data. The Dp16 is a partial trisomy and therefore it cannot be assumed that phenotypic features observed in it will be observed in full trisomy. Discussion of this point needs to be included.

            Some specific details:

Line 69: what is “euthanasia solution”? Was it used for all experiments? Because this is not a rapid procedure (i.e. unlike cervical dislocation) and because there is no mention of tissues being placed on ice or the time to get tissues into a solution that would preserve RNA stability and inhibit protein degradation, it must be assumed that signifcant changes in RNA/protein profiles were occurring, and potentially degradation. Note that this is different from a simple RNA quality assessment used in RNA sequencing protocol. This lowers the confidence in measurements of RNA/protein levels.

Hemodynamics: were mice sacrificed after these experiments? How many mice of each genotype and sex were used?

            Fulton score (lines 117 and 120): Are these the same mice as used for RNA sequencing? How were these sacrificed? There are apparently two sets of mice here; state numbers of mice of each genotype and sex.

            Similar information needs to be provided for the Flexivent and Elisa experiments, and for immunoblots and immunoflourescence, and histomorphometry (which clearly used a different set of mice). A table (replacing the rather useless (and poorly titled) current Table 1) would be easiest, one that lists each experiment with its associated number of mice (genotype and sex), method of sacrifice and age, with an indication of which experiments used the same cohorts of mice. There should also be a statement of which mice are littermates and how many litters were represented in each experiment and between experiments; dam/sire information also would be ideal. This latter information could go in the Supplementary material. It is best practice to not use e.g. a single litter for any individual experiment. With an n of three, a single litter could have been used.

Results

Line 237: “no differences were noted in either sex between controls and Dp16 mice.” This makes no sense. Not least because the next sentence lists genotype differences.

            Table 1. What are these numbers? The title states “General characteristics” which is clearly not what this table shows. Are these the total number of mice used in all experiments? Are the %s indicating % male and female? Why are there twice as many controls as Dp16? (note that totals of male and female are useless numbers when control and Dp16 are added together)  Replace the table as suggested above to provide numbers of mice used in each experiment, and provide clear labels.

            In Figure 1, separate graphs by sex. It is inadequate to say there are no sex differences; the reader needs to see the data.

            Figure 1b, why is the number of mice so low compared to data in Fig1a? Again separate by sex.

            Line 250, legend to Figure 1: “Hydrocephalus was not observed in control mice but occurred at highest frequency in female Dp16 mice.”. These data are not shown in the Figure???

Figure 2: Clarify the title – are these RNA or protein data and how determined and from where? Data need to be separated by sex. Note that there are known sex differences in gene expression; the assumption cannot be made that there are none for the RNA/proteins chosen for measurement here. An n of 3 means the data are unreliable. Comment on the stunning variance in Dp16, in particular for Il4, Il6, Il10, etc. Is this caused, at least in part, by the very low n?

Table 2: the data presented are completely inadequate and cannot be evaluated. No information is provided on analysis techniques, quality control, p values, correction for multiple testing, etc. Provide actual numerical differences, fold change, % difference, not merely increased or decreased. And errors. Include all Hsa21 orthologs detected, trisomic and disomic; you do not know that only the Ifnar genes are relevant (if there are large numbers of Hsa21 orthologs with detectable expression, these can be provided in a separate Supplementary Table). Were expression levels of any orthologs of other Hsa21 genes, those mapping to Mmu17 and Mmu10, altered?

            Figure 3: in the graph labels, change “expression” to either mRNA or protein. How many mice were used in each measurement? If n=3, that will contribute to the very high variances seen and makes the conclusions invalid.

            Table 3: Change “expression” to protein. Again, provide quantitative data, i.e. %increase etc, with error measurement

            Table 4: separate by sex

            Figure 5: state the n for each set. Large variances preclude conclusion with very small n

Discussion

            Line 476: the authors appear troubled that genes encoding many of the protein/RNA abberations that they measured are not trisomic. Logically, given that the Mmu16 trisomic segment encodes transcription factors, kinases, protein and RNA modifiers/degradation regulators, etc, if expression levels are perturbed, disruptions among non trisomic genes is a very logical expectation. Note too that not all DS carry an extra chromosome (e.g. Robertsonian translocation etc). Modify the discussion accordingly.

            Line 491: The authors also seem troubled by the lack of correlation between RNA and protein levels. This is not new; see e.g Schwanhuaser B et al 2011, Maier et al 2009, PMID 27104977.

            Line 502: More circumspect interpretation of inter-individual variation in protein levels is required. The number of mice is as small as 3. This is not sufficient. In addition, with insufficient care in methods of euthanasia and sample preservation (as seems to have occurred), some amount of RNA/protein degradation/alteration will have occurred. Thus, no conclusions can be made regarding this important question, although the authors’ statements regarding clinical interventions are well taken.

            Line 551 and discussion of CBS. CBS is not trisomic in the Dp16. This point needs to be made and some discussion provided regarding how results here might be influenced by trisomy of CBS as would occur in most people with DS.

            As an extension of the CBS question, these results need to be put in context of the limited trisomy that the Dp16 mice provide. This is of critical importance. The Dp16 is a partial trisomy; only 60% of Hsa21 orthologous protein coding genes are trisomic. Results obtained with it thus cannot be assumed to represent those in full Hsa21 trisomy Down syndrome. Unless of course the other 40% of Hsa21 genes (i.e. 60 protein coding genes) have no functiosn relevant to lung or heart. Some discussion should be provided regarding which of the ~40% of Hsa21 orthologs that are not trisomic in the Dp16 might influence any of the measurements. E,g, the authors reference PCP4 regarding cilia formation; what about PCNT, also demonstrated to affect cilia development? What other genes are known or might be predicted to affect lung or heart function or immune responses? TRPM2, PDE9A, endostatin, etc. This cannot be answered with great certainty or thoroughness, but some effort must be made, and the reasonableness of the question needs to be acknowledged. What data exist regarding relevant phenotypes of other mouse models, e.g. those trisomic for Mmu10 and Mmu17 orthologous segments, and/or those models crossed with either the Ts65Dn or the Dp16? The reader needs to understand the limitations of the analysis here in order to think clearly about next steps. Note that this does not diminish the importance and usefulness of the data here. However, it helps to prevent the assumption, so popular in the DS field, that data from the Dp16 or the Ts65Dn can be directly extrapolated to human DS. Which they absolutely cannot. And leads to clinical trials that have uniformly failed to produce significant practical results.

Other points:

            Line 82: refer to the Dp16 mice consistently as Dp16 not, as here, as simply “Dp”

            Line 180, immunoblots “Antibodies against beta actin were used to probe blots and estimate loading”; Figure 3c is labelled as GADPH

            Line 202: clarify “Seven Dp16 (three males/ four females) and seven Dp16-

(controls, three males and four females)”. Dp16 should not be listed twice

            Be consistent in using bold when referring to figures

            In each figure legend, state the number of mice in each group (age would also be helpful)

            Everywhere change “expression” to either RNA or protein, as appropriate

            Line 516-517: “..cells and tissues can directly attenuate Hsa21 gene (over) dosage”: add “consequences of” .. (over) dosage. i.e. cells and tissues cannot get rid of the extra Hsa21

            Correct background strain: C57Bl/6J. Also, note that for awhile The Jackson Lab was breeding the Dp16 onto F1 hybrids of C57Bl/6JEi X C3H, the same as for the Ts65Dn. It was not clear in the Methods section if the mice used here were directly purchased from JAX or bred in house from stocks obtained from JAX years ago. Either way, please verify and provide the correct background strain name.

Author Response

On behalf of my co-authors, we sincerely thank the reviewers for their careful evaluation of our original manuscript entitled “Lung Heart Biology of the Dp16 Mouse Model of Down syndrome: Implications for Studying Cardiopulmonary Disease”. We have provided a "marked up" copy and a "clean version of the revised manuscript. We have taken great care and time to review the concerns raised by the reviewers. In addressing these important issues, we hope the reviewers agree that the manuscript is now clearer and more polished. The following is a point-by-point set of responses specifically addressing each of the three reviewers’ comments, concerns, suggestions, and questions.

Reviewer 1

We thank the reviewer for providing a careful, meticulous and passionate critique of our work.

General Comments and Suggestions:

  1. The reviewer cites flawed design of the RNA seq experiments due to having too few mice, with 3 males and 3 females each from controls and from Dp16 mice.

Response: Our goal for the paper was to provide an overview of lung heart biology in the Dp16 mice. As such, we were not asking questions specific to sex differences. We are certainly aware of sex differences in RNA quantification studies. We followed the recommendations of sample size set forth by Schurch et al in their paper in RNA in 2016 (PMID 27022035), in which they concluded that 6 biological replicates are sufficient for fold change calls of 2.0 or higher comparing two groups using the majority of available RNA-seq differential gene expression tools and statistical tests. We cite this paper now in the RNA seq methods section. Nevertheless, we fully agree with the reviewer that the sample size of 6 v 6 increases the rate of a type II error (false negative). At P = 0.65 for alpha 0.05, our study has a 35% likeliness of a type II error. We struck a balance between cost and practicality with lower statistical power. The mice are on a highly inbred strain, and we are only highlighting genes with large fold changes.

  1. The reviewer cites a lack of consistent information on the number of mice used in each experiment.

Response: We agree and we have added numbers to all figures and figure legends.

  1. The reviewer indicates that the manuscript was not carefully prepared and needs a complete inspection and overhaul for professional manuscript preparation.

Response: We agree and this has been done.

  1. The reviewer requests that discussion needs to be included reflecting that Dp16 is a partial trisomy and therefore phenotypic features may not fully correspond to human trisomy.

Response: We agree and it is now included in the discussion.

Specific Details:

  1. Line 69: what is euthanasia solution? RNA stability concerns.

Response: All mice were sacrificed expeditiously, and all samples placed on ice and preserved appropriately, as we have published previously. We have updated the methods section to reflect this and we now include detailed concentration data on pentobarbital in the euthanasia solution.

2.Hemodynamics: were mice sacrificed after these experiments?

Response: Yes, this is a terminal procedure. We have now included the number of mice for each genotype and sex for this and all experimental endpoints throughout.

  1. Fulton scores: are these the same mice used for RNA sequencing?

Response: No, the process of weighing the sections of hearts from mice precludes their use for molecular analyses such as RNA or protein work. We have now included the number of mice for each genotype and sex for this and all experimental endpoints throughout.

  1. The reviewer suggests using a table to represent which mice and how many were used for each set of experiments.

Response: This has now been provided.

Results

  1. Line 237 makes no sense.

Response: This has now been corrected.

  1. Table 1: what are these numbers?

Response: We disagree with the reviewer about poor readability of the table and agree with the other 2 reviewers as well as other articles by ourselves and many others which use a nearly identical table format. Nevertheless, we replaced the table as suggested.

  1. In Figure 1, separate graphs by sex. It is inadequate to say there are no sex differences.

Response: The reviewer misinterpreted the statement. The statement does not include the word phenotype or a related term. The statement clearly refers to sex, i.e. male or female (we purposefully avoided use of the term “gender”). As expected, we found no statistical difference in the number of male vs female offspring from either control mice or Dp16 mice. We have changed the language in the text to reflect this. We also have modified Figure 1 to depict males and females.

  1. In Figure 1b, why is the number of mice so low compared to data in Figure 1a?

Response: These data correspond to a subset of mice that were anesthetized for the Flexivent study. We have updated the text accordingly. No sex data are provided because we did not power this study to examine whether there were differences in response to anesthesia between males and females; it was powered for control vs. Dp16, as is the case for the entire project at this time. Future studies will examine sex more closely.

9.Line 250, legend to Figure 1: Hydrocephalus was not observed… These data are not shown in the figure.

Response: We have removed this language from the figure legend.

  1. Figure 2: Clarify the title- are these RNA or protein data and how determined and from where?... An n of 3 means the data are unreliable. Comment on the stunning variance in Dp16.. in part by the low n?

Response: The text preceding Figure 2 clearly states blood plasma, the methods state by Mesoscale Discovery Platform, and the Figure 2 Legend states that mice “circulate” a unique profile of cytokines and chemokines and also states plasma. Nevertheless, we have added the words “blood plasma” into the figure legend title.

We have reconfigured the graph to include sex differences, and this is now presented as Supplemental Figure 1. However, we politely disagree with the reviewer that an n=3 is unreliable. We would point the reviewer to numerous studies with an n=3 for males and for females each  for two comparison groups. The Mesoscale platform, as is the case for other multiple ELISA platforms, is very expensive. We, as many other research groups have done and continue to do, struck a balance between n numbers and high cost. We felt this was appropriate because our primary focus is differences between controls and Dp16 not differences due to sex.

With regard to stunning variance in Dp16, we point the reviewer to dozens of studies on humans with DS and in mouse models of DS that present blood plasma ELISA data. For each analyte we measured, the published literature range spans an order of magnitude. Put simply, there is no consensus on what a “reference range” for blood plasma cytokines and chemokines is in DS or mouse model of DS. We have provided references strongly supporting this conclusion. Rather than toss out high or low values as outliers, we elected to include all data. We have since removed samples from our analyses if their values were more than 2 standard deviations away from the mean, and then we re-analyzed the data (and separated the data by sex).

  1. Table 2: the data presented are completely inadequate and cannot be evaluated…

Response: We reformatted the table as suggested. We also added language in the methods describing RNA seq analyses, including differentially expressed genes fold call change, p value, etc., using DESeq2.

  1. Figure 3: in the graph labels, change expression to mRNA or protein. How many mice were used in each measurement?

Response: We changed the axis labels as suggested and provided n numbers.

  1. Table 3: Change expression to protein… provide quantitative data with error measurement

Response: This has been done per suggestion.

  1. Table 4: Separate by sex

Response: The data analyses from the invasive hemodynamics suite is complex, and re-performing the analyses by sex would take a significant amount of time. Currently, Jules Harral, our co-author, is the only person who can perform this, and she is booked with catheterizations for several weeks in the future. Thus, we did not perform this request from the reviewer.

  1. Figure 5: state the n for each set.

Response: This has been done per suggestion.

Discussion

  1. Line 476: the authors appear troubled that genes encoding…Modify the discussion accordingly.

Response: The reviewer misunderstood the statement. The discussion is speaking to the finding that blood plasma cytokines and chemokine are different in DS and DS mouse models compared to typical individuals. We make the important statement that none of the measured cytokines are encoded on chromosome 16, so that the reader can be reminded that there expression differences are not the result of gene duplication. Line 476 states exactly what the reviewer is suggesting, that other transcription factors, RNA modifiers, etc. are perturbed and so differences are a very logical expectation. We have added a sentence to the discussion that highlights our agreement with the reviewer.

  1. Line 491: the authors also seem troubled by the lack of correlation between RNA and protein levels.

Response: On the contrary, we agree with the reviewer and that was our point. We made this point in the Discussion because currently, the field of DS basic research is dominated by RNA-centric studies, with much less effort made to study at the protein level. This aspect of our discussion is a reminder to the field that some current “dogma” is based heavily (almost exclusively in some papers) on RNA work. Given the differences between RNA and protein expression, which we are certainly aware of and agree with the reviewer, we hope to caution the DS research community about drawing conclusions from studies which lack protein data.

  1. Line 502: more circumspect interpretation of inter-individual variation in protein levels is required… with insufficient care in methods of euthanasia and sample preservation (as seems to have occurred)…

Response: We have included n numbers throughout, indicating that only the RNA seq data is 3 vs 3. Other recent publications have observed the inter individual variability we are reporting. We have added this language as requested and also included these references. We have added more detailed information into the methods section regarding the euthanasia solution and placing samples on ice. We have also added the following to the methods section: Pentobarbital with secondary thoracotomy is strongly preferred over cervical dislocation or carbon dioxide exsanguination. This technique allows for the heart and lungs to continue to work and to prevent clotting while blood is collected. Additionally, lung and heart tissues are flushed with saline before being removed and placed on ice. This allows for high quality histology particularly for vascular structures; flushing is not available with other euthanasia techniques, and cervical dislocation stops lung function immediately.

  1. Line 551 and discussion of CBS… these results need to be put in context of the limited trisomy that the Dp16 mice provide.

Response: We agree and this language has been added and/or modified per the reviewers suggestions.

20.Line 82: refer to the Dp16 mice consistently.

Response: We have made sure this is now consistent throughout.

  1. Line 180: beta actin.. gapdh..

Response: “Beta actin” has been corrected to GAPDH.

  1. Line 202: Seven Dp16 and seven Dp16-

Response: “Dp16-” has been corrected to controls.

  1. Be consistent in using bold. In each figure legend, state the number of mice in each group. Everywhere change “Expression” to either mRNA or protein.

Response: All of these have been corrected as suggested.

  1. Line 516-517: add “consequences of”

Response: Corrected as suggested

  1. Correct background strain: C57Bl/6J.

Response: Corrected as suggested

Reviewer 2 Report

The authors investigated the baseline lung and heart biology of the 12 Dp16 mouse model of Down syndrome (DS) compared to controls using analytes from blood samples. This is an interesting and well conducted study. Just a few minor comments to improve the manuscript. 

1) You may consider rephrase the title since the current version is a bit confusing, especially the "Lung Heart Biology". "Lung and Heart Biology may be more clear" 

2) For abstract,  I would suggest add one sentence regarding the significance of this work - Why you want to investigated the baseline lung and heart biology and why it is important? 

3) It is not clear what specifically "lung sample" refers to, is it lung tissue or lung pleura? Should be more specific and accurate. 

4) I would suggest change the presentation of Table 1. You seem care more about sex difference rather than case vs. control? I would suggest change the column to DP status. 

Author Response

Reviewer 2

  1. Consider rephrasing the title..

Response: Corrected as suggested

  1. Consider adding one sentence of significance to the abstract…

Response: Corrected as suggested

  1. Unclear what lung sample means…

Response: We have corrected this throughout to refer to lung parenchyma or blood plasma.

  1. Change the presentation of Table 1.

Response: Corrected as suggested

Reviewer 3 Report

NICE JOB CONGRATS!FEW THINGS TO TAKE UNDER CONSIDERATION:

1)THE ABSTRACT IS DIFFICULT TO READ,BE MORE CLEAR WITH LESS DETAILS

2)INTRODUCTION SHOULD BE CORRECTED  GRAMMATICALLY(LINE 49)OR SCIENTIFICALLY(L.56)FOR EXAMPLE.

3)IN RESULTS/DISCUSSION TRY TO BE MORE STRAITFORWARD WITH SMALLER SENTENCES AND PARAGRAFS

Author Response

Reviewer 3

  1. Abstract needs to be more clear with less details…

Response: Corrected as suggested

  1. Introduction should be corrected grammatically and scientifically…

Response: Corrected as suggested

  1. Results and discussion need to be more straightforward with smaller sentences and paragraphs

Response: Corrected as suggested
